

**Investigation of the relationship between electrical conductivity (EC) of**
**water and soil, and landform classes using fuzzy model and GIS**
**Marzieh Mokarram[1] and Dinesh Sathyamoorthy[2]**
[1]*Marzieh Mokarram(Department of Range and Watershed Management, College of Agriculture*
*and Natural Resources of Darab, Shiraz University, Iran, Email: m.mokarram@shirazu.ac.ir)*
[2]*Dinesh Sathyamoorthy (Science & Technology Research Institute for Defence (STRIDE),*
*Ministry of Defence, Malaysia (E-mail: dinesh.sathyamoorthy@stride.gov.my)*
***Corresponding author:*** *Marzieh Mokarram, Tel.: +98-917-8020115; Fax: +987153546476 ,*
*Address: Darab, Shiraz university, Iran, Postal Code: 71946-84471, Email:*
*m.mokarram@shirazu.ac.ir*



**Abstract**

In this research, the relationship between classes of landform, and electrical conductivity (EC)

of soil and water in the Shiraz Plain, Fars province Iran was investigated using a combination

of geographical information system (GIS) and fuzzy model. The results of the fuzzy method for

water EC showed that 36.6% of the land to be moderately land suitable for agriculture; high,

31.69%; and very high, 31.65%. In comparison, the results of the fuzzy method for soil EC showed

that 24.31% of the land to be as not suitable for agriculture (low class); moderate, 11.78%; high,

25.74%; and very high, 38.16 %. In the total, the land suitable for agriculture with low EC is

located in the north and northeast of the study area. The relationship between landform and EC

shows that EC of water is high for the valley classes, while EC of soil is high in the upland drainage

class. In addition, the lowest EC for soil and water are in the plain small class.

**Keywords:** Groundwater quality, landform, electrical conductivity (EC), fuzzy model.

36

## 1. Introduction

Soil features are largely controlled by the landforms on which they are developed. The

physiographic penetration on soil properties is recognized based on the progress of the soil–





landform relationship (Ali and Moghanm, 2013). According to landform formed by the same

geomorphic processes, it is the main key of feature because it can easily be identified, and it is also

that were responsible for making the undercoat material of the soils (Park and Burt, 2002;

Henderson et al., 2005; Mini et al. 2007; Poelking et al., 2015). Also the research show that there

is a clear relationship between landform and soils. So that the soil and the landforms control the

hydrological erosional, biological, and geochemical cycles and based on type of landform can be

predicted other parameters of watershed such as soil, erosion, biological and so on (Berendse et

al., 2015; Brevik et al., 2015; Decock et al., 2015; Keesstra et al., 2012; Smith et al., 2015)

Usage of remote sensing and geography information system (GIS) enable the production of multi

presentive layers of soil properties, which provide a great source of data for the land use planners

(Ali et al., 2007).

GIS, with features like the ability to acquire and exchange many different sources, organization,

retrieval and display of data, analysis of numerous data, and possibility to provide multiple

services, has been introduced as an efficient tool in the planning. Combining GIS with fuzzy logic

provides a comparatively new land evaluation method (Badenki and kurtener, 2004; Oinam et al,

2014; Wang et al., 2015). Incorporating both of these methods is more flexible, and reflects human

creativeness and understanding more and more to make decisions. Fuzzy inference is considered

as a deduction for mathematical modeling in imprecise and vague processes, uncertainty about

data and thus makes a context for modeling uncertainly (Kurtener, 2005).

Ali and Moghanm (2013) studied the variation of soil properties over the landforms around Idku

Lake, Egypt. The spatial distribution of $CaCO_3$, EC, organic matter (OM), pH, nitrogen (N),

phosphor (P), potassium (K), iron (Fe), manganese (Mn), copper (Cu) and zinc (Zn) over the

various landforms was discussed in detail. The results show that the change of $CaCO_3$, EC and OM





is minimal in the landforms of sand sheets, hammocks, sabkhas, clay flats and former lake-bed.
Aliabadi and Soltanifard (2014) apply GIS and fuzzy inference for determination of the impact
of water and soil EC, and calcium carbonate on wheat crop. Regarding the results of the fuzzy
inference system, 76% was achieved using the of Mamdani and 52 percent of accuracy for the
technique Sugeno were achieved.
Also by El-Keblawy et al (2015) investigated relationships between landforms, soil characteristics
and dominant xerophytes in the northern United Arab Emirates. Soil texture, electrical
conductivity (EC) and pH were determined in each stand.
Also the results show that the soil and the landforms also control the geomorphological and
hydrological processes (Cerdà and García-Fayos, 1997, Cerdà, 1998, Dai et al, 2015, Nadal-
Romero et al., 2015).
One of the largest wheat producing regions was located in the Shiraz Plain, Fars province Iran
(Bijanzadeh et al., 2014). The aim of this study is to investigate of the relationship between
landform classes and EC of water and soil in the Shiraz Plain using a combination of GIS and
fuzzy model. The methodology employed in this study is summarized in Figure 1.





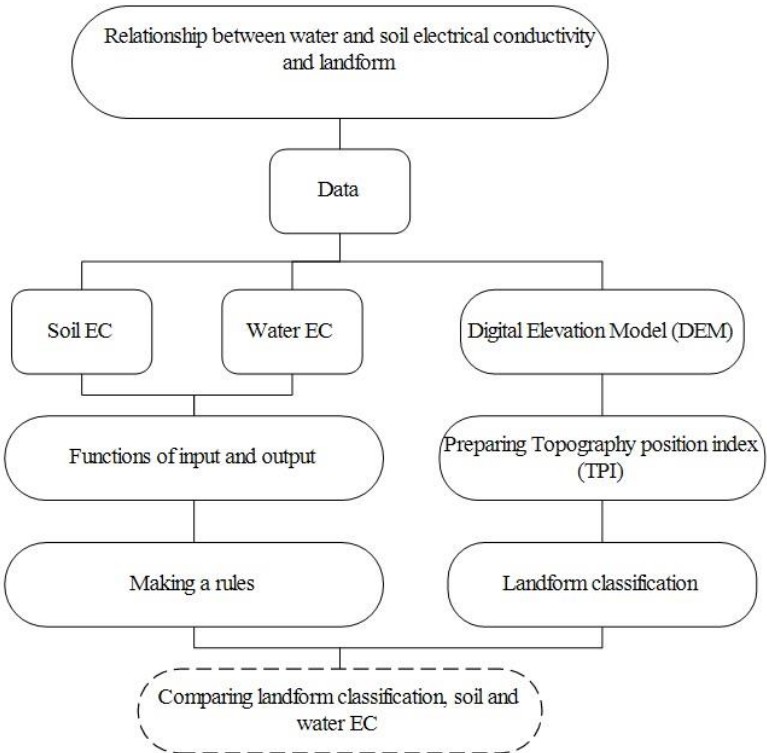


Figure 1. Flowchart of the methodology employed to investigate the relationship between landform
classification, and soil and water EC.

**2. Case study**
The study area has an area of 3,909 km$^2$ and is located at longitude of N 29° 06′- 29° 43′and
latitude of E 52° 18′ to 53° 28′ (Figure 2). The altitude of the study area ranges from the lowest
of 1,433 m to the highest of 3,083 m. The region is located in the north of the Fars province, which
has cold winters with hot summers. The average temperature for the area is 16.8 °C, ranging
between 4.7 and 29.2 °C (Soufi, 2004). The research area is a biodiversity of mountains, relief and
lithology, and geological characteristics such as for instance sedimentary basin and elevated reliefs

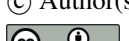



(Soufi, 2004). The main land use types of the region are agriculture, range land, farming and
forests.

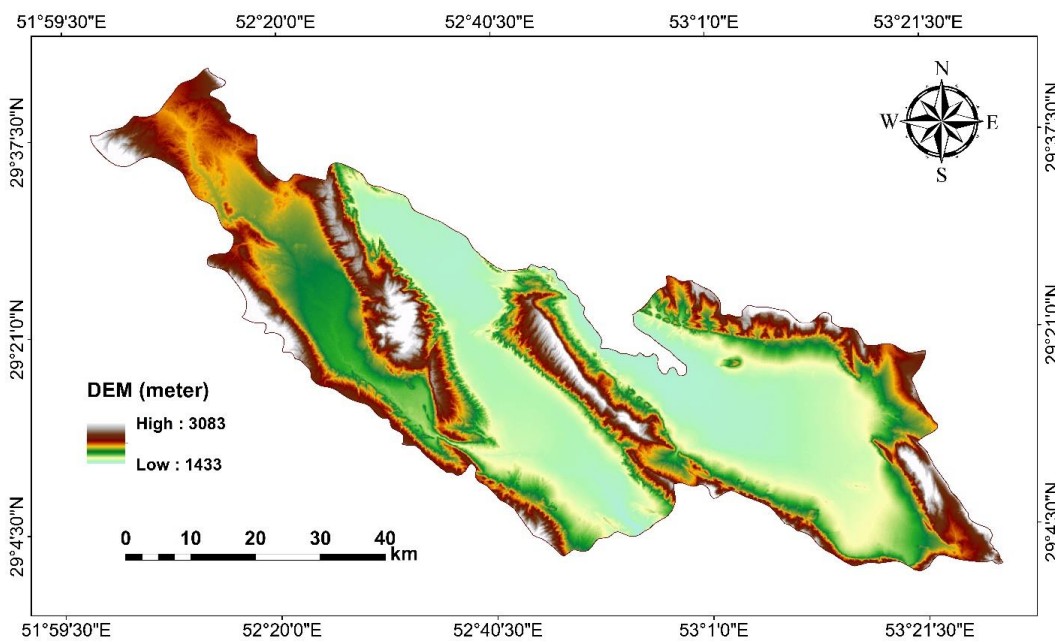


Figure 2. Location of the study area (DEM with spatial resolution of 30 m) (Source:
http://earthexplorer.usgs.gov/).

The evaluation of land suitability for agricultural production (in particular wheat crop) in the area
is essentialist critical, and should consider environmental factors and human conditions (Soufi,
2004; Bijanzadeh et al., 2014).One of the factors that is main in the amount of soil and water
salinity.

**3. Materials and methods**



**3.1. Inverse Distance Weighted (IDW)**

IDW model was used for interpolating the EC properties. IDW interpolation explicitly implements the assumption that things that are close to one another are more alike than those that are farther apart. To predict a value for any unmeasured location, IDW will be used that measures neighborhood values in the predicted location. Assumed value of an attribute $f$ at any unsampled point is an average of distance-weighted of sampled points lying within a defined neighborhood around that unsampled point. Basically it is a weighted moving average (Burrough, et al., 1998):

$$\hat{f}(x_0) = \frac{\sum_{i=1}^{n} f(x_i) d_{ij}^{-r}}{\sum_{i=1}^{n} d_{ij}^{-r}} \tag{1}$$

Where $x_0$ is the estimation point and $x_i$ are the data points within a chosen surrounding. The weights ($r$) are related to distance by $d_{ij}$.

**3.2. Fuzzy method**

In the research, model functions are accustomed to compute membership function (MF), as described in Figure 3 (Burrough and McDonnell, 1998). In such status, an asymmetric function needs to be applied (Models 1 and 2) (Figure 3). If $MF(x_i)$ shows individual membership value for $i^{th}$ land property $x$, then in the computation process these model functions (Models 1 to 2) show the following form:

For *asymmetric left* (Model 1):





$MF(x_i) = [1/(1 + \{(x_i - a_i - b_1)/b_1\}^2)] \, if \, x_i < (a_1 + b_1)$         (2)

For asymmetric right (Model 2):
$MF(x_i) = [1/(1 + \{(x_i - a_2 + b_2)/b_2\}^2)] \, if \, x_i > (a_2 - b_2)$         (3)

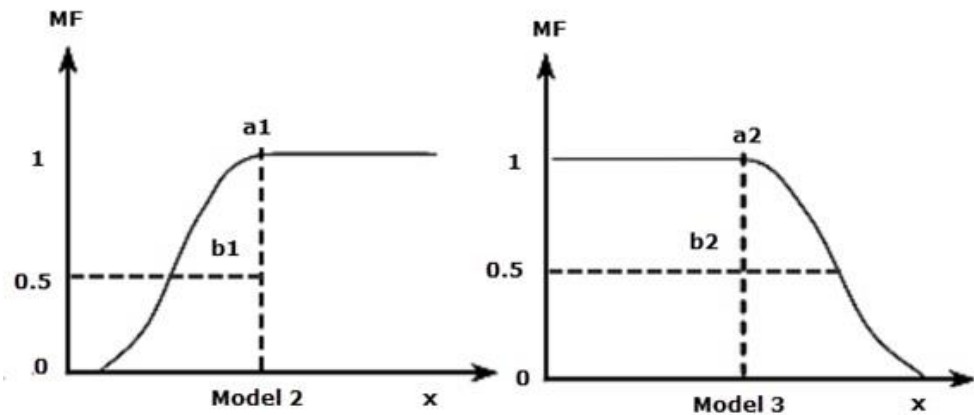


Figure 3. Membership functions.

In this study, in order to define fuzzy rule based membership functions, the categories shown in
Tables 1 and 2 are used.

Table 1. Classification of water EC values (Kumar et al., 2003).

| Class | EC (ds/m) |
|---|---|
| Low | $< 0.25$ |
| Moderate | $0.25 - 0.75$ |
| High | $0.75 - 2.25$ |
| Very high | $> 2.25$ |








Table 2. Classification of soil EC values (Mokarram et al., 2010).

| Class | EC (ds/m) |
|---|---|
| Low | < 8 |
| Moderate | 8-12 |
| High | 12-16 |
| Very high | > 16 |



**3.3. Landform classification**

TPI (Weiss, 2001) compares the elevation of each cell in a DEM to the mean elevation of a specified
neighborhood around that cell. Positive
TPI (Eq. (4)) compares the elevation of each cell in a DEM to the mean elevation of a defined
neighborhood around that cell. Mean elevation is subtracted from the elevation value at center
(Weiss 2001):
$$TPI_i = Z_0 - \frac{\sum_{n-1} Z_n}{n} \qquad\qquad (4)$$
where;
$Z_0$= elevation of the model point under evaluation
$Z_n$= elevation of grid
$n$ = the total number of surrounding points employed in the evaluation

Incorporating TPI at small and large scales permit a number of nested landforms to be distinguished
(Table 3). The actual breakpoints among classes can be selected to optimize the classification for a
specific landscape. As in slope position classifications, additional topographic metrics, such as for



example differences of elevation, slope, or aspect within the neighborhoods, can help delineate
landforms more accurately (Weiss 2001).
Table 3.Topographic Position Index (TPI) thresholds for small and large neighborhoods used to
define landscape feature classes

| Landform | TPI | |
|---|---|---|
| Landform | Small Neighborhood | Large Neighborhood |
| Plains | -1 < TPI < 1 | -1<TPI<1* |
| Open slopes | -1 < TPI < 1 | -1<TPI<1** |
| U-shaped valleys | -1 < TPI < 1 | TPI < -1 |
| Mountain tops/High ridges | TPI > 1 | TPI > 1 |
| Upper slopes/Mesas | -1 < TPI < 1 | TPI > 1 |
| Midslope drainages/Shallow valleys | TPI < -1 | -1 < TPI < 1 |
| Canyons/Deeply incised streams | TPI < -1 | TPI < -1 |
| Midslope ridges/Small hills in plains | TPI > 1 | -1 < TPI < 1 |
| Upland drainages/Headwaters | TPI < -1 | TPI > 1 |
| Local ridges/Hills in valleys | TPI > 1 | TPI < -1 |
| *Plain landform class required a slope of < 0.5 | | |
| **Open slopes landform class required a slope of > 0.5 | | |


Also the classes of canyons, deeply incised streams, midslope and upland drainages, shallow
valleys, and tend to have strongly negative plane form curvature values. On the other hand, local
ridges / hills in valleys, midslope ridges, small hills in plains and mountain tops, and high ridges
have strongly positive plane form curvature values.


**4. Results and Discussion**
**4.1. Inverse Distance Weighted (IDW)**
IDW interpolation was used to produce the prediction of soil and water EC, as shown in Figure
4. The lowest and highest output for IDW were 0.016 and 14.48 respectively for water EC, while
the lowest and highest soil EC were 0 and 34.5 respectively. The interpolation maps for soil and



water EC are shown in Figure 5. The statistical properties of the interpolated soil and water EC
are shown in Table 4.

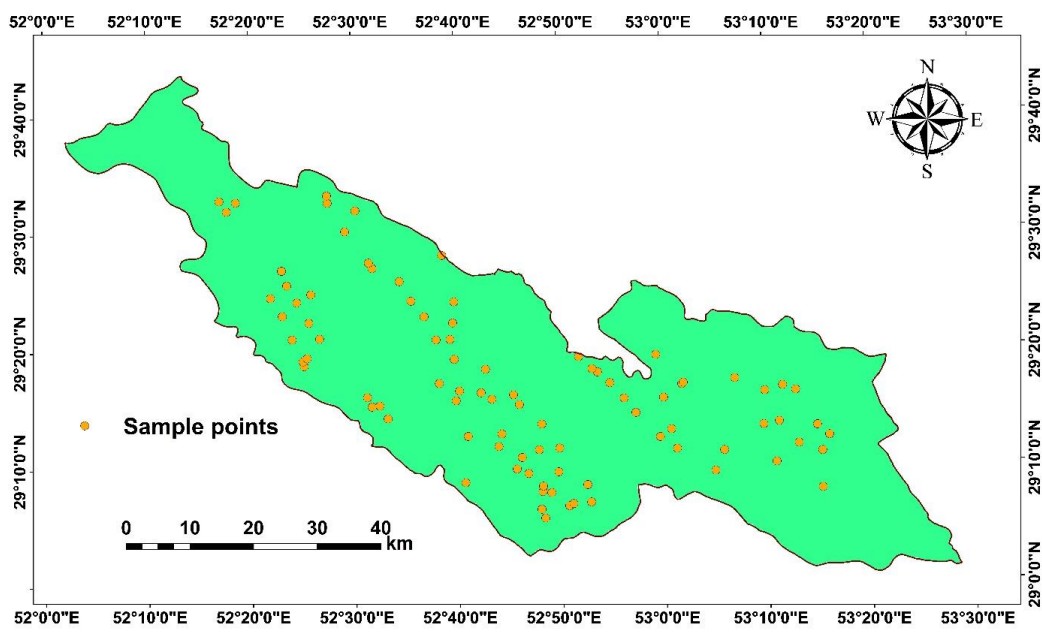

(a)



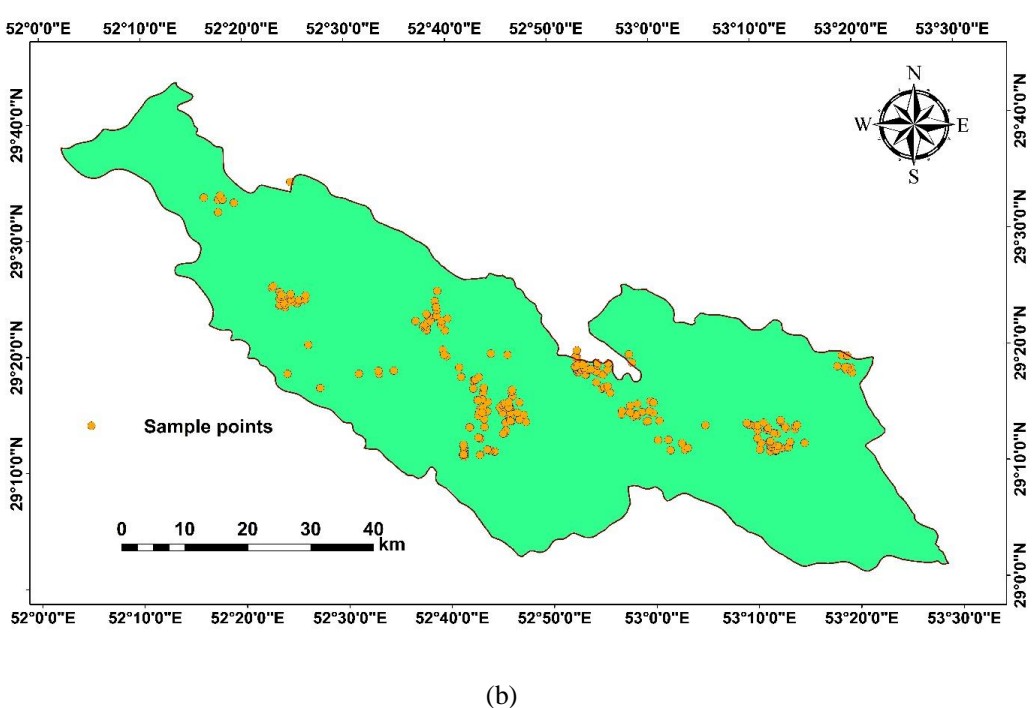

(b)

Figure 4. Position of sample points for (a) water and (b) soil EC.

Table 4. Descriptive statistics of the EC water and EC soil

| Statistic parameter | EC water (ds/m) | EC soil (ds/m) |
|---|---|---|
| Maximum | 14.48 | 28.25 |
| Minimum | 0.016 | 0.78 |
| Average | 3.80 | 3.91 |
| STDEV | 6.13 | 3.82 |
| Skewness | 6.54 | 3.09 |
| Kurtosis | 62.97 | 15.46 |





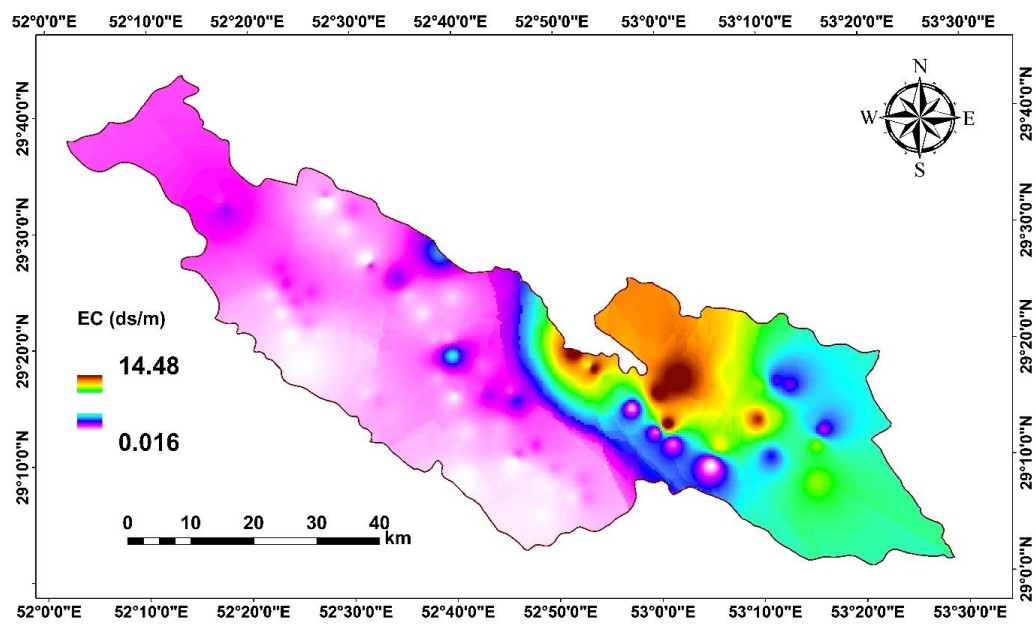

(a)

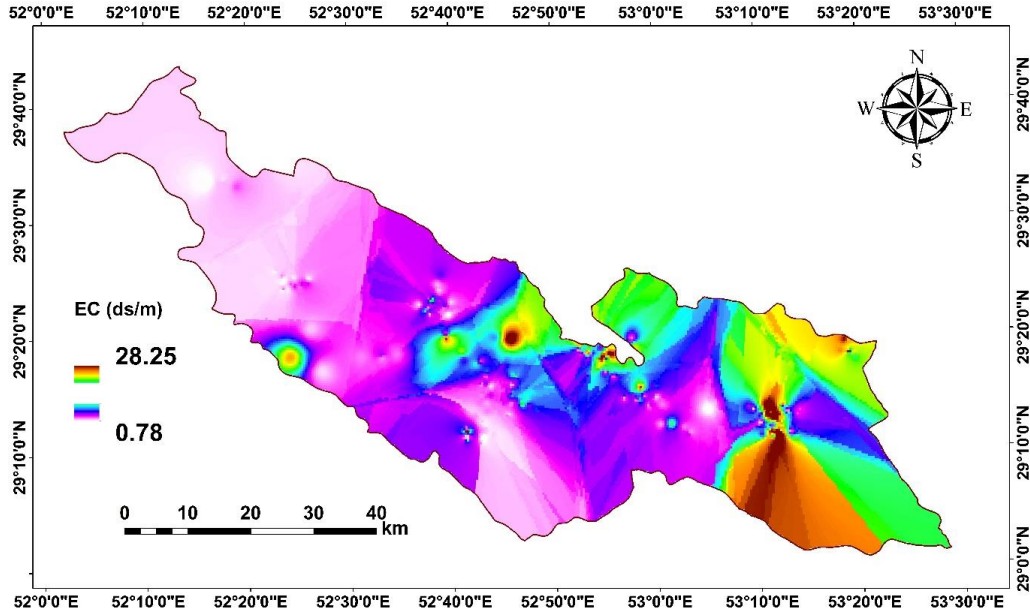

(a)





Figure 5. Interpolated maps of study area for (a) water and (b) soil EC.

**4.2. Fuzzy method**
Fuzzy maps were prepared for soil and water EC, as shown in Figure 6. The fuzzy values were
classified into four classes. EC < 0.25, EC between 0.25-0.5, EC between 0. 5-0.75 and EC > 0.75
are in the classes of low, moderate, high and very high respectively (Shobha et al., 2014). The
areas of the classes for soil and water EC are shown in Table 5.

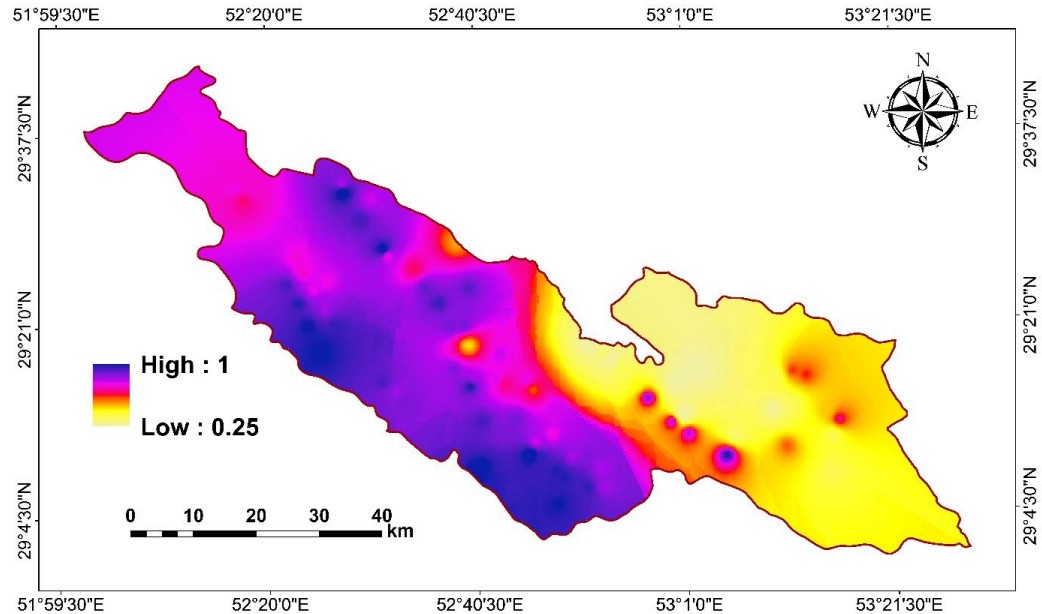

(a)




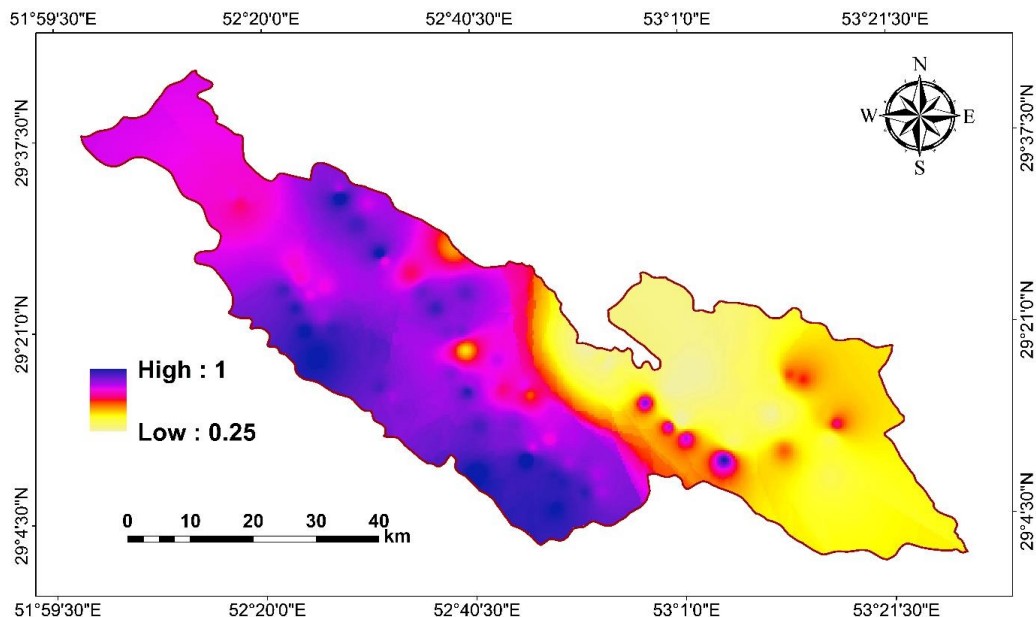

(b)

Figure 6. Fuzzy maps of the study area for (a) soil and (b) water EC.

Table 5. Areas of the classes for water and soil EC.

| Class | Area (%) | | Area (km$^2$) | |
|---|---|---|---|---|
| | Water EC | Soil EC | Water EC | Soil EC |
| Low | 0.00 | 24.31 | 0.11 | 950.23 |
| Moderate | 36.60 | 11.78 | 1430.87 | 460.63 |
| High | 31.69 | 25.74 | 1238.91 | 1006.27 |
| Very high | 31.65 | 38.16 | 1237.10 | 1491.86 |

For water EC, the fuzzy model showed that 36.6% of the land was in the moderate class; high, 31.69%; and very high, 31.65%. In comparison, the results of the fuzzy model for soil EC showed that 24.31% of the land was in the low class; moderate, 11.78%; high, 25.74%; and very high, 38.16 %. Based on the results obtained, the land suitable for wheat agriculture is located in the north and northeast in the study area.





**4.3. Landform classification**
In order to determine of relationship between landform classification, and soil and water EC, the
landform map of the study area was prepared. Using TPI, the landform classification map of the
study area was generated. The TPI maps generated using small and large neighborhoods are shown
in Figures 7. TPI is between -106 to 130 and -334 to 533 for 3 and 45 cells for small and large
neighborhoods respectively (Figure 8). The landform maps generated based on the TPI values are
shown in Figure 8. The classification has ten classes; high ridges, midslope ridges, upland
drainage, upper slopes, open slopes, plains, valleys, local ridges, midslope drainage and streams.
The areas of the landform classes are shown in Figure 9. It is observed that the largest landform is
streams, while the smallest is plains.

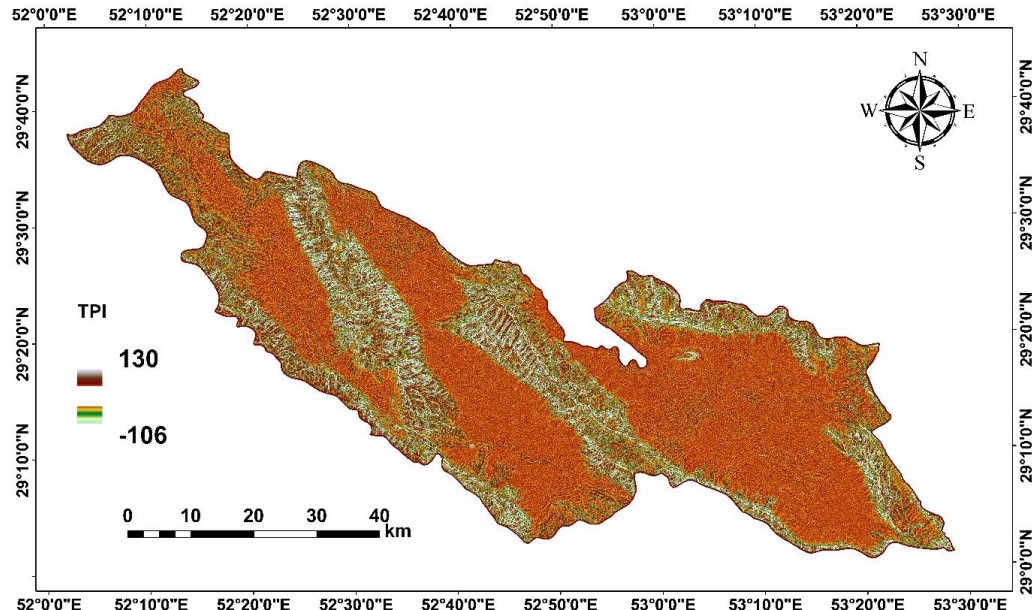

(a)



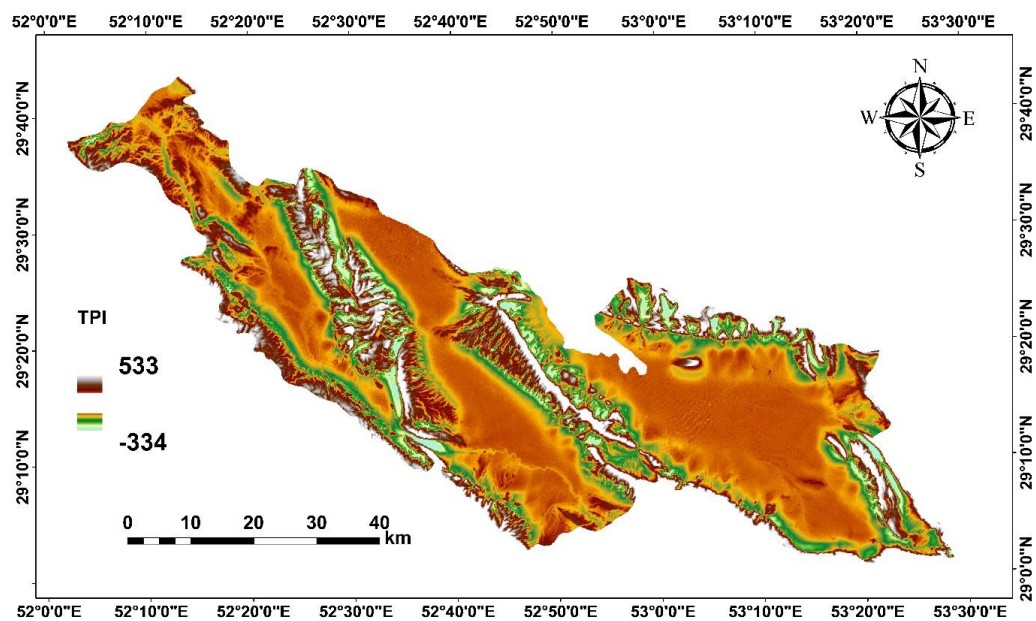

(b)

Figure 7. TPI maps generated using (a) small (3 cells) and (b) large (45 cells) neighborhood.





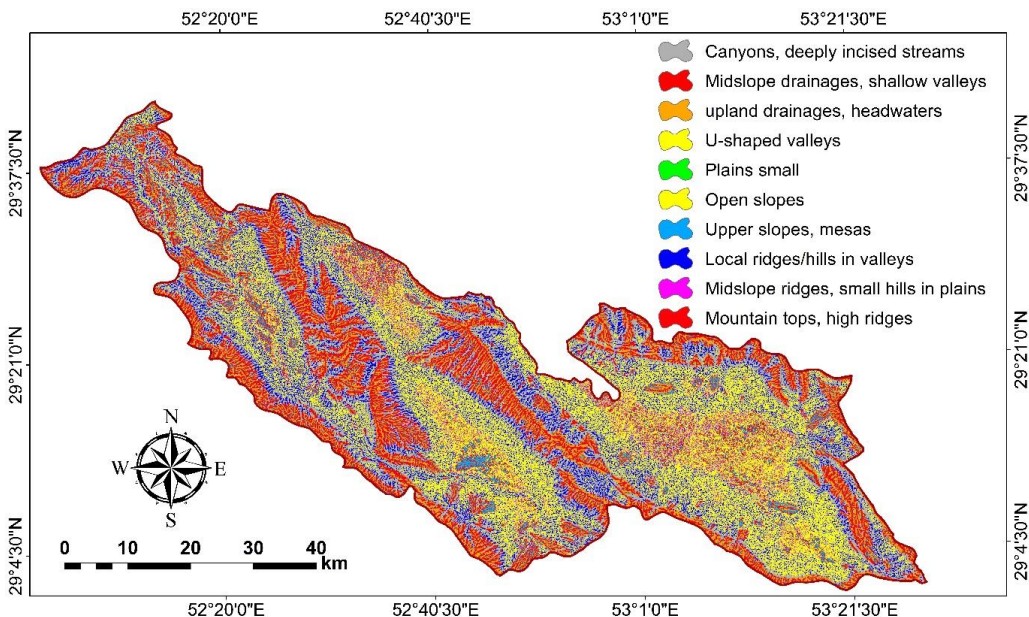

Figure 8. Landform classification using the TPI method.

The average EC for each landform class was determined, and the relationship between EC and landform was prepared. According to Figure 9, EC of water is high for the valley class while the high EC of soil is in upland drainage class. The lowest EC for soil and water are in the plain small class.








Figure 9. Relationship between landform classes.

Ali and Moghanm (2013), who investigated relationship between soil properties and landform
classes in Idku Lake, Egypt, also found that the lowest EC was in plain class. In fact there is
relationship between soil parameters and land use (Wasak and Drewnik, 2015; Debasish-Saha et
al., 2014). Yu et al. (2012) showed that there is relationship between soil parameters (such as soil
organic carbon (SOC), soil total nitrogen (STN)) and types of land cover (grassland, farmland,
swampland, ….). Niu et al. (2015) and Yu et al. (2015) investigated relationship between land use
and soil moisture. The results provided an insight into the significances for land use and farming
water management in this area. Even the studies show that there is relationship between soil
structural stability and land use (Saha and Kukal, 2015). The results showed that a degradation of
soil physical attributes due to the conversion of natural ecosystems to farming system and increased erosion
hazards in the lower. So the soil parameters depend with land use, so that with changes in land use, they
also change.

**5. Conclusion**



In this study, the relationship between classes of landform, and electrical conductivity (EC) of soil and water was in the Shiraz Plain was investigated using a combination of geographical information system (GIS) and fuzzy model. The results of the fuzzy method for water EC showed that 36.6% of the land to be moderately land suitable for agriculture; high, 31.69%; and very high, 31.65%. In comparison, the results of the fuzzy method for soil EC showed that 24.31% of the land to be as not suitable for agriculture (low class); moderate, 11.78%; high, 25.74%; and very high, 38.16 %. In the total, the land suitable for agriculture with low EC is located in the north and northeast of the study area. The relationship between landform and EC shows that EC of water is high for the valley classes, while EC of soil is high in the upland drainage class. In addition, the lowest EC for soil and water are in the plain small class.

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
