# Peer review of "1. Introduction"

_Solid Earth, 2016_

## Referee Comment (RC1) · Dr Oliva (Referee) · 24 Mar 2016

This paper complements several ongoing studies of the authors about soil fertility in Iran. This study, together with some recent submitted papers, focuses one of the most important centres for agriculture in this country. Consequently, a better understanding of the factors influencing soil fertility in this region may provide important guidelines to improve agriculture production for the entire country.

I have started reviewing the manuscript until the results section. Until here, I had some minor comments and suggestions, which you will find below. But when I reached the

[Figure]

Result and Discussion section I found it very difficult to follow. It is very poor and mainly consists of figures, without accurate descriptions of their meaning. Moreover, and this is the weakest part of the manuscript, results are not discussed and compared with other similar studies and regions on Earth. This section needs to be rewritten and reorganized. Please, interpret and discuss your data, not just present it. What is new and different with respect to previous studies? Are similar/different approaches with similar/different results been implemented in other areas with similar environmental settings? Please support your results comparing your data with other similar studies around the world.

Figures and tables are enough and of good quality.

Title The Study area is not included and should be since this is a study case focusing on one specific area.

Abstract l. 25 as a geomorphologist, it is not clear to me what "classes of landforms" means

Keywords: one keyword should be related to the study area

Introduction l. 40-42 unclear, please rephrase l. 43-44 Repeating what was said before l. 44-46 too forced, please split the two ideas into two different sentences l. 51-53 this is a scientific paper, please be direct and do not mention this kind of general information. People that will read your findings already know what GIS possibilites are. Avoid unnecessary information. l. 56 "more and more"??? l. 59-60 and 60-62. Merge these two sentences. l. 68 and 71 avoid same constructions in consecutive sections l. 71 "also" appears two times in the same sentence

2. Case study The Study area should include further information for the reader who is not familiar with the area (geology, water availability, human population/distribution, etc). l. 86 what about precipitation? I guess it is important for agriculture purposes. l. 87 monthly or absolute values? I guess it is monthly, mention it l. 92 is there any

important city/village in the study area which should be included in the map? l. 96 human conditions? Mention them. Is the area under great human pressure? l. 97 space before "one" l. 97-98, this sentence is uncomplete. I would say, OK, and what? Please rephrase it.

---

## Author Comment (AC1) · 2 Apr 2016

Reply to Reviewer's Comments

Title The Study area is not included and should be since this is a study case focusing on one specific area. Keywords: one keyword should be related to the study area The study area has been added to the titles and keywords.

Abstract l. 25 as a geomorphologist, it is not clear to me what "classes of landforms" means. It has been replaced by landform classification.

[Figure]

Introduction: 40-42 unclear, please rephrase l. This sentence has been corrected.

43-44 Repeating what was said before This sentence has been corrected.

44-46 too forced, please split the two ideas into two different sentences This sentence has been split two different sentences.

51-53 this is a scientific paper, please be direct and do not mention this kind of general information. People that will read your findings already know what GIS possibilities are. Avoid unnecessary information. This sentence has been removed.

l. 56 "more and more"??? This mistake has been removed.

59-60 and 60-62. Merge these two sentences. 68 and 71 avoid same constructions in consecutive sections . 71 "also" appears two times in the same sentence These amendments have been made.

Case study The Study area should include further information for the reader who is not familiar with the area (geology, water availability, human population/distribution, etc). These more information about case study has been added.

86 what about precipitation? I guess it is important for agriculture purposes. The information about rainfall has been added.

87 monthly or absolute values? I guess it is monthly, mention it. The monthly is correction that has been added.

92 is there any important city/village in the study area which should be included in the map? Figure 2 has been corrected.

human conditions? Mention them. Is the area under great human pressure? The information about it, I could not find.

97 space before "one" l. 97-98, this sentence is uncomplete. I would say, OK, and what? Please rephrase it. It was mistake that has been removed.

Result and Discussion section I found it very difficult to follow. It is very poor and mainly consists of figures, without accurate descriptions of their meaning. Moreover, and this is the weakest part of the manuscript, results are not discussed and compared with other similar studies and regions on Earth. This section needs to be rewritten and reorganized. Please, interpret and discuss your data, not just present it. What is new and different with respect to previous studies? Are similar/different approaches with similar/different results been implemented in other areas with similar environmental settings? Please support your results comparing your data with other similar studies around the world. It has been improved.

Please also note the supplement to this comment:
http://www.solid-earth-discuss.net/se-2016-30/se-2016-30-AC1-supplement.zip

––––––––––––––––––––––––––––––––

---

## Author Comment (AC2) · 2 Apr 2016

Reply to Reviewer's Comments

Title The Study area is not included and should be since this is a study case focusing on one specific area. Keywords: one keyword should be related to the study area The study area has been added to the titles and keywords.

Abstract l. 25 as a geomorphologist, it is not clear to me what "classes of landforms" means. It has been replaced by landform classification.

[Figure]

Introduction: 40-42 unclear, please rephrase l. This sentence has been corrected.

43-44 Repeating what was said before This sentence has been corrected.

44-46 too forced, please split the two ideas into two different sentences This sentence has been split two different sentences.

51-53 this is a scientific paper, please be direct and do not mention this kind of general information. People that will read your findings already know what GIS possibilities are. Avoid unnecessary information. This sentence has been removed.

l. 56 "more and more"??? This mistake has been removed.

59-60 and 60-62. Merge these two sentences. 68 and 71 avoid same constructions in consecutive sections . 71 "also" appears two times in the same sentence These amendments have been made.

Case study The Study area should include further information for the reader who is not familiar with the area (geology, water availability, human population/distribution, etc). These more information about case study has been added.

what about precipitation? I guess it is important for agriculture purposes. The information about rainfall has been added.

monthly or absolute values? I guess it is monthly, mention it. The monthly is correction that has been added.

is there any important city/village in the study area which should be included in the map? Figure 2 has been corrected.

human conditions? Mention them. Is the area under great human pressure? The information about it, I could not find.

space before "one" l. 97-98, this sentence is uncomplete. I would say, OK, and what? Please rephrase it. It was mistake that has been removed.

Result and Discussion section I found it very difficult to follow. It is very poor and mainly consists of figures, without accurate descriptions of their meaning. Moreover, and this is the weakest part of the manuscript, results are not discussed and compared with other similar studies and regions on Earth. This section needs to be rewritten and reorganized. Please, interpret and discuss your data, not just present it. What is new and different with respect to previous studies? Are similar/different approaches with similar/different results been implemented in other areas with similar environmental settings? Please support your results comparing your data with other similar studies around the world. It has been improved.

Please also note the supplement to this comment:
http://www.solid-earth-discuss.net/se-2016-30/se-2016-30-AC2-supplement.pdf

**Supplement:**

**Investigation of the relationship between electrical conductivity (EC) of water and soil, and landform classification in the northern part of Meharloo watershed, Fars province, Iran using fuzzy model and GIS**

**Marzieh Mokarram[1] and Dinesh Sathyamoorthy[2]**

[1]*Marzieh Mokarram(Department of Range and Watershed Management, College of Agriculture and Natural Resources of Darab, Shiraz University, Iran, Email: m.mokarram@shirazu.ac.ir)*

[2]*Dinesh Sathyamoorthy (Science & Technology Research Institute for Defence (STRIDE), Ministry of Defence, Malaysia (E-mail: dinesh.sathyamoorthy@stride.gov.my)*

***Corresponding author:*** *Marzieh Mokarram, Tel.: +98-917-8020115; Fax: +987153546476 , Address: Darab, Shiraz university, Iran, Postal Code: 71946-84471, Email: m.mokarram@shirazu.ac.ir*

**Investigation of the relationship between electrical conductivity (EC) of water and soil, and landform classification in the northern part of Meharloo watershed, Fars province, Iran using fuzzy model and GIS**

**Abstract**

In this research, the relationship between landform classification and electrical conductivity (EC) of soil and water in the in the northern part of Meharloo watershed, Fars province,

Iran was investigated using a combination of geographical information system (GIS) and fuzzy model. The results of the fuzzy method for water EC showed that 36.6% of the land to be moderately land suitable for agriculture; high, 31.69%; and very high, 31.65%. In comparison, the results of the fuzzy method for soil EC showed that 24.31% of the land to be as not suitable for agriculture (low class); moderate, 11.78%; high, 25.74%; and very high, 38.16 %. In the total, the land suitable for agriculture with low EC is located in the north and northeast of the study area. The relationship between landform and EC shows that EC of water is high for the valley classes, while the EC of soil is high in the upland drainage class. In addition, the lowest

EC for soil and water are in the plain small class.

**Keywords:** Meharloo watershed, Groundwater quality, landform, electrical conductivity (EC), fuzzy model.

**1. Introduction**

Soil features are largely controlled by the landforms on which they are developed. The physiographic penetration on soil properties is recognized based on the progress of the soil–landform relationship (Ali and Moghanm, 2013). The landforms formed by the same geomorphic processes is the main key feature because they can easily be identified, and were responsible for making the undercoat material of the soils (Park and Burt, 2002; Henderson et al., 2005; Mini et al. 2007; Poelking et al., 2015). Previous studies have shown that there is a clear relationship between landform and soils, in that landforms and soil both control hydrological erosional, biological, and geochemical cycles. Based on the type of landform, other parameters of watersheds can be predicted, such as soil, erosion, biological and so on (Berendse et al., 2015; Brevik et al., 2015; Decock et al., 2015; Keesstra et al., 2012; Smith et al., 2015)

Geographical information systems(GIS) GIS, with features such as the ability to acquire and exchange many different sources, organization, retrieval and display of data, analysis of numerous data, and possibility to provide multiple services, has been introduced as an efficient tool in the planning. Combining GIS with fuzzy logic provides a comparatively new land evaluation method (Badenki and Kurtener, 2004; Oinam et al, 2014; Wang et al., 2015). Incorporating both of these methods is more flexible, and reflects human creativeness and understanding to make decisions. Fuzzy inference is considered as a deduction for mathematical modeling in imprecise and vague processes, uncertainty about data and thus makes a context for modeling uncertainly (Kurtener, 2005).

Ali and Moghanm (2013) studied the variation of soil properties over the landforms around Idku Lake, Egypt, with tthe spatial distribution of $CaCO_3$, EC, organic matter (OM), pH, nitrogen (N), phosphor (P), potassium (K), iron (Fe), manganese (Mn), copper (Cu) and zinc (Zn) over the various landforms discussed in detail. The results showed that the changes of $CaCO_3$, EC and

OM are minimal in the landforms of sand sheets, hammocks, sabkhas, clay flats and former lake- bed.

Aliabadi and Soltanifard (2014) apply GIS and fuzzy inference for determination of the impact of water and soil EC, and calcium carbonate on wheat crop. Regarding the results of the fuzzy inference system, 76% was achieved using the of Mamdani and 52% of accuracy for the Sugeno technique was achieved.

In addition, El-Keblawy et al (2015) investigated relationships between landforms, soil characteristics and dominant xerophytes in the northern United Arab Emirates. Soil texture, electrical conductivity (EC) and pH were determined in each stand. The results showed that soil and landforms also control the geomorphological and hydrological processes (Cerdà and García-

Fayos, 1997, Cerdà, 1998, Dai et al, 2015, Nadal-Romero et al., 2015).

One of the largest wheat producing regions in Iran is located in the Shiraz Plain, Fars province (Bijanzadeh et al., 2014). The aim of this study is to investigate of the relationship between landform classes and EC of water and soil in this area using a combination of GIS and fuzzy models. The methodology employed in this study is summarized in Figure 1.

[Figure]

Figure 1. Flowchart of the methodology employed to investigate the relationship between landform classification, and soil and water EC.

**2. Case study**

The study area has an area of 3,909 km$^2$ and is located at longitude of N 29° 06′- 29° 43′and latitude of E 52° 18′ to 53° 28′ (Figure 2). The altitude of the study area ranges from the lowest of 1,433 m to the highest of 3,083 m. The region is located in the north of the Fars province, which has cold winters with hot summers. The average temperature for the area is 16.8 °C, ranging between 4.7 and 29.2 °C (Soufi, 2004). The research area is a biodiversity of mountains, relief and lithology, and geological characteristics such as for instance sedimentary basin and elevated reliefs (Soufi, 2004). The main land use types of the region are agriculture, range land, farming and forests.

In terms of geology  the Precambrian Hormoz series and the Quaternary units are the oldest and youngest rocks in the basin, respectively. Spans of outcropped rocks, covering from the

Cretaceous to Quaternary, are carbonate sediments of deep to shallow marine facies. These sedimentary sequences include large and small stratigraphic gaps in the form of disconformity and sometimes nonconformity (Khaksar et al., 2006).

The area is situated in an arid and semi-arid region. Rainfall varies from 150mm on the plains to

650mm on the high mountains, with an average of 350 mm. The rainfall is concentrated in cold seasons, while the precipitation is very low from June to October (Sigaroodi et al., 2014).

During winter, several migratory bird species from north of Caspian Sea, flamingos (Phoenicopterus roseus), common shelducks (Tadorna tadorna) and mallards (Anas platyrhynchos), spend 4 months in the area feeding on brine shrimp (Artemia franciscana). Thus, the lake has important ecological value (Sigaroodi et al., 2014).

[revised manuscript text omitted]

29. Sigaroodi, S. K. Chen, Q. Ebrahimi, S. Nazari, A. and Choobin B.: Long-term
precipitation forecast for drought relief using atmospheric circulation factors: a study
on the Maharloo Basin in Iran. Hydrol. Earth Syst. Sci., 18, 1995–2006.2014.

30. Singh, D.P., and Rathore, M.S.: Morphological, physical and chemical properties of
soils associated in top sequence for establishing taxonomy classes in Pratapgarh

District of Rajasthan, India. African Journal of Agricultural Research.  Vol. 10(25), pp. 2516-2531, 18 June, 2015

31. Smith, P., Cotrufo, M. F., Rumpel, C., Paustian, K., Kuikman, P. J., Elliott, J. A., McDowell, R., Griffiths, R. I., Asakawa, S., Bustamante, M., House, J. I., Sobocká, J., Harper, R., Pan, G., West, P. C., Gerber, J. S.,  Clark, J.M., Adhya, T., Scholes, R.J. and Scholes, M.C.: Biogeochemical cycles and biodiversity as key drivers of ecosystem services provided by soils. Soil 1, 665-685, DOI:10.5194/soil-1-665-2015.

32. Soufi, M.: Morpho-climatic classification of gullies in fars province, southwest of i.r. iran . International Soil Conservation Organisation Conference – Brisbane, 2004.

33. Walia, C.S. and Chamuah, G.S.: Soils of riverine plain in Arunachal plain and their suitability for some agricultural crops. J. Indian Soc. Soil Sci. 42:425-429. 1994.

34. Wang, J., Ge, A., Hu, Y., Li, C. and Wang, L.: A fuzzy intelligent system for land consolidation - A case study in Shunde, China Solid Earth, 6 (3), pp. 997-1006, DOI: 10.5194/se-6-997-2015.

35. Wasak, K. and Drewnik, M.: Land use effects on soil organic carbon sequestration in calcareous Leptosols in former pastureland-a case study from the Tatra Mountains (Poland). Solid Earth, 6 (4), pp. 1103-1115, 2015.

36. Weiss, A.: Topographic Positions and Landforms Analysis (Conference Poster). ESRI, 2001.

37. Yu, B., Stott, P., Di, X.Y. and Yu, H. X.: Assessment of land cover changes and their effect on soil organic carbon and soil total nitrogen in daqing prefecture, China. Land Degradation and Development, 25 (6), pp. 520-531, 2014.

38. Yu,Y., Wei, W., Chen, L. D., Jia, F. Y., Yang, L., Zhang, H. D. and Feng, T. J.:

   Responses of vertical soil moisture to rainfall pulses and land uses in a typical loess

   hilly area, China. Solid Earth, 6 (2), pp. 595-608. Cited 1 time, 2015.

---

## Referee Comment (RC2) · M. Shaygan (Referee) · 16 Apr 2016

The paper has an interesting idea and and focuses on an important field in agriculture and soil studies. Anyway it needs some revisions which is as following: The introduction is well written and is fine. but about the study area it is better to show a full map which show the location of the study area in country and give some more information about it..spatially the lands and things like that

The structure of sections is fine.but about the method the sample points are not spreaded well, please tell us how the sample points are collected?

quality of maps are fine and it is good.But there is some spelling mistakes which should be reviewed

with kind regards

---

## Author Comment (AC3) · 16 Apr 2016

Author's Response

1- About the study area it is better to show a full map which show the location of the study area in country and give some more information about it. Spatially the lands and things like that In page 7, line 114 was used the study area with more details. Also was inserted agriculture properties in page 6, line 91-92.

2- About the method the sample points are not separated well, please tell us how the sample points are collected? In page 11, line 197-198 was explained about sample

points.

Please also note the supplement to this comment:
http://www.solid-earth-discuss.net/se-2016-30/se-2016-30-AC3-supplement.pdf

**Supplement:**

| ١  | Investigation of the relationship between electrical conductivity (EC) of                          |
|----|----------------------------------------------------------------------------------------------------|
| ۲  | water and soil, and landform classes in the northern part of Meharloo                              |
| ٣  | watershed, Fars province, Iran using fuzzy model and GIS                                           |
| ٤  | Marzieh Mokarram 1 and Dinesh Sathyamoorthy 2                                |
| 0  |                                                                                                    |
| ٦  | 1 Marzieh Mokarram(Department of Range and Watershed Management, College of Agriculture |
| ٧  | and Natural Resources of Darab, Shiraz University, Iran, Email: m.mokarram@shirazu.ac.ir )  |
| ٨  | 2 Dinesh Sathyamoorthy (Science & Technology Research Institute for Defence (STRIDE),   |
| ٩  | Ministry of Defence, Malaysia (E-mail: dinesh.sathyamoorthy@stride.gov.my)                         |
| ۱. | Corresponding author: Marzieh Mokarram, Tel.: +98-917-8020115; Fax: +987153546476,          |
| ۱۱ | Address: Darab, Shiraz university, Iran, Postal Code: 71946-84471, Email:                          |
| ١٢ | m.mokarram@shirazu.ac.ir                                                                    |
| ۱۳ |                                                                                                    |
| ١٤ |                                                                                                    |
| 10 |                                                                                                    |
| ١٦ |                                                                                                    |
| ١٧ |                                                                                                    |
| ١٨ |                                                                                                    |
| ۱۹ |                                                                                                    |

**۲۱**

Investigation of the relationship between electrical conductivity (EC) of
 water and soil, and landform classes in the northern part of Meharloo
 watershed, Fars province, Iran using fuzzy model and GIS

**Abstract**

۲۷ In this research, the relationship between landform classification and electrical conductivity ۲۸ (EC) of soil and water in the in the northern part of Meharloo watershed, Fars province, Iran ۲٩ was investigated using a combination of geographical information system (GIS) and fuzzy model. ۳. The results of the fuzzy method for water EC showed that 36.6% of the land to be moderately ۳١ land suitable for agriculture; high, 31.69%; and very high, 31.65%. In comparison, the results of ٣٢ the fuzzy method for soil EC showed that 24.31% of the land to be as not suitable for agriculture ٣٣ (low class); moderate, 11.78%; high, 25.74%; and very high, 38.16%. In the total, the land suitable ٣٤ for agriculture with low EC is located in the north and northeast of the study area. The relationship ۳0 between landform and EC shows that EC of water is high for the valley classes, while the EC of ٣٦ soil is high in the upland drainage class. In addition, the lowest EC for soil and water are in the ۳۷ plain small class.

Keywords: Meharloo watershed, Groundwater quality, landform, electrical conductivity (EC),
fuzzy model.

**٤) **1. Introduction**

٤٢ Soil features are largely controlled by the landforms on which they are developed. The ٤٣ physiographic penetration on soil properties is recognized based on the progress of the soil-٤٤ landform relationship (Ali and Moghanm, 2013). The landforms formed by the same geomorphic 20 processes is the main key feature because they can easily be identified, and were responsible for ٤٦ making the undercoat material of the soils (Park and Burt, 2002; Henderson et al., 2005; Mini et ٤٧ al. 2007; Poelking et al., 2015). Previous studies have shown that there is a clear relationship ٤٨ between landform and soils, in that landforms and soil both control hydrological erosional, ٤٩ biological, and geochemical cycles. Based on the type of landform, other parameters of watersheds ٥. can be predicted, such as soil, erosion, biological and so on (Berendse et al., 2015; Brevik et al., 01 2015; Decock et al., 2015; Keesstra et al., 2012; Smith et al., 2015)

٥٢ Geographical information systems(GIS) GIS, with features such as the ability to acquire and ٥٣ exchange many different sources, organization, retrieval and display of data, analysis of numerous 0 2 data, and possibility to provide multiple services, has been introduced as an efficient tool in the 00 planning. Combining GIS with fuzzy logic provides a comparatively new land evaluation method ٥٦ (Badenki and Kurtener, 2004; Oinam et al, 2014; Wang et al., 2015). Incorporating both of these ٥٧ methods is more flexible, and reflects human creativeness and understanding to make decisions. ٥٨ Fuzzy inference is considered as a deduction for mathematical modeling in imprecise and vague 09 processes, uncertainty about data and thus makes a context for modeling uncertainly (Kurtener, ٦. 2005).

Ali and Moghanm (2013) studied the variation of soil properties over the landforms around Idku Lake, Egypt, with the spatial distribution of CaCO3, EC, organic matter (OM), pH, nitrogen (N), phosphor (P), potassium (K), iron (Fe), manganese (Mn), copper (Cu) and zinc (Zn) over the various landforms discussed in detail. The results showed that the changes of CaCO3, EC and OM
 are minimal in the landforms of sand sheets, hammocks, sabkhas, clay flats and former lake-bed.
 Aliabadi and Soltanifard (2014) apply GIS and fuzzy inference for determination of the impact
 of water and soil EC, and calcium carbonate on wheat crop. Regarding the results of the fuzzy
 inference system, 76% was achieved using the of Mamdani and 52% of accuracy for the Sugeno
 technique was achieved.

In addition, El-Keblawy et al (2015) investigated relationships between landforms, soil
 characteristics and dominant xerophytes in the northern United Arab Emirates. Soil texture,
 electrical conductivity (EC) and pH were determined in each stand. The results showed that soil
 and landforms also control the geomorphological and hydrological processes (Cerdà and García Fayos, 1997, Cerdà, 1998, Dai et al, 2015, Nadal-Romero et al., 2015).

۷٥

One of the largest wheat producing regions in Iran is located in the Shiraz Plain, Fars province
 (Bijanzadeh et al., 2014). The aim of this study is to investigate of the relationship between
 landform classes and EC of water and soil in this area using a combination of GIS and fuzzy
 models. The methodology employed in this study is summarized in Figure 1.

A) Figure 1. Flowchart of the methodology employed to investigate the relationship between landform

At classification, and soil and water EC.

۸۳

٨.

**$\wedge \epsilon$ **2.** Case study**

The study area has an area of 3,909 km2 and is located at longitude of N 29° 06′- 29° 43′ and latitude of E 52° 18′ to 53° 28′ (Figure 2). The altitude of the study area ranges from the lowest of 1,433 m to the highest of 3,083 m. The region is located in the north of the Fars province, which has cold winters with hot summers. The average temperature for the area is 16.8 °C, ranging between 4.7 and 29.2 °C (Soufi, 2004). The research area is a biodiversity of mountains, relief and lithology, and geological characteristics such as for instance sedimentary basin and elevated reliefs (Soufi, 2004). The main agricultural produce consists of grain, fruit, and vegetables, while the
 partly wooded mountains are used for pasture. The main land use types of the region are
 agriculture, range land, farming and forests.

In terms of geology the Precambrian Hormoz series and the Quaternary units are the oldest and
 youngest rocks in the basin, respectively. Spans of outcropped rocks, covering from the Cretaceous
 to Quaternary, are carbonate sediments of deep to shallow marine facies. These sedimentary
 sequences include large and small stratigraphic gaps in the form of disconformity and sometimes
 nonconformity (Khaksar et al., 2006).

The area is situated in an arid and semi-arid region. Rainfall varies from 150mm on the plains to
650mm on the high mountains, with an average of 350 mm. The rainfall is concentrated in cold
seasons, while the precipitation is very low from June to October (Sigaroodi et al., 2014).

During winter, several migratory bird species from north of Caspian Sea, flamingos
 (Phoenicopterus roseus), common shelducks (Tadorna tadorna) and mallards (Anas
 platyrhynchos), spend 4 months in the area feeding on brine shrimp (Artemia franciscana). Thus,
 the lake has important ecological value (Sigaroodi et al., 2014).

- ١٠٦
- ۱۰۷
- ١٠٨
- 1.9
- 11.
- 111

۱۱۲